# Effect of the Graphene Quantum Dot Content on the Thermal, Dynamic-Mechanical, and Morphological Properties of Epoxy Resin

**DOI:** 10.3390/polym15234531

**Published:** 2023-11-25

**Authors:** Bárbara Schneider, Heitor Luiz Ornaghi Jr., Francisco Maciel Monticeli, Daiane Romanzini

**Affiliations:** 1Mantova Indústria de Tubos Plásticos Ltd.a., Rua Archimedes Manenti, 574, B. Centenário, Caxias do Sul 950450-175, RS, Brazil; bscheider.89@gmail.com; 2Postgraduate Program in Technology and Materials Engineering (PPGTEM), Federal Institute of Education, Science and Technology of Rio Grande do Sul (IFRS), R. Princesa Isabel, 60, Feliz 95770-000, RS, Brazil; ornaghijr.heitor@gmail.com (H.L.O.J.); daiane.romanzini@feliz.ifrs.edu.br (D.R.); 3Department of Aerospace Structures and Materials, Delft University of Technology, Mekelweg 5, 2628 CD Delft, The Netherlands

**Keywords:** carbon quantum dots, epoxy nanocomposite, thermal conductivity, structure and property relationship

## Abstract

Different amounts of graphene quantum dots (CQDs) (0, 1, 2.5, and 5 wt%) were incorporated into an epoxy matrix. The thermal conductivity, density, morphology, and dynamic mechanical thermal (DMTA) properties were reused from the study of Seibert et al.. The Pearson plot showed a high correlation between mass loading, thermal conductivity, and thermal diffusivity. A poorer correlation with density and heat capacity was observed. At lower CQD concentrations (0.1 wt%), the fracture surface showed to be more heterogeneous, while at higher amounts (2.5 and 5 wt%), a more homogeneous surface was observed. The storage modulus values did not change with the CQD amount. But the extension of the glassy plateau increased with higher CQD contents, with an increase of ~40 °C for the 5 wt% compared to the 2.5 wt% and almost twice compared to the neat epoxy. This result is attributed to the intrinsic characteristics of the filler. Additionally, lower energy dissipation and a higher glass transition temperature were observed with the CQD amount. The novelty and importance are related to the fact that for more rigid matrices (corroborated with the literature), the mechanical properties did not change, because the polymer bridging mechanism was not present, in spite of the excellent CQD dispersion as well as the filler amount. On the other hand, thermal conductivity is directly related to particle size and dispersion.

## 1. Introduction

Nanomaterials are materials with size ranging from 1 to 100 nm [1] (one nanometer is equivalent to ten H atoms). Many properties are achieved due to their high surface area, which greatly differs from their bulk counterparts, giving distinct magnetic, electrical, mechanical, optical, and other properties. Different nanomaterials, such as fullerenes, carbon nanotubes, nanodiamonds, nanoporous materials, core–shell nanoparticles, and carbon quantum dots, find their role in scratch-free paints, surface coatings, electronics, cosmetics, environmental remediation, and sensors, among other things [1]. For the production of the nanomaterials, there are two different approaches: the top-down approach (mechanical milling, etching, laser ablation, sputtering, electro-explosion) or the bottom-up approach (supercritical fluid synthesis, spinning, sol-gel approach, laser pyrolysis, chemical vapor deposition, molecular condensation, chemical reduction, and green synthesis) [2]. The dependent effects are more prominent at the nanoscale than at the meso- or macroscale. For example, the mechanical properties of nanomaterials are higher compared to their bulk counterparts due to the increase in crystal perfection or reduction in crystallographic defects, but the stress transfer mechanism also changes. For example, the polymer bridging mechanism mainly causes the tensile deformation of materials containing nanoparticles, where the deformation is avoided even if the interfacial interaction is excellent [3]. For bulk materials, more specifically fibers, if the interfacial interaction is excellent, the tensile property will be higher than the neat matrix because the fiber (more rigid than the matrix) receives all the stress applied and retains most of the stress received.

Carbon-based quantum dots (graphene quantum dots and carbon quantum dots) are the shining stars of nanomaterials and present a vast possibility of applications (as can be seen in the schematic representation of Figure 1) [4]. The constant advance of technology and science allows the number of applications to increase even more. Scientists know even more about the strategies (size, modification, synthetic methods) to produce the expected properties (optical, luminescent) for biomedicine, optronics, catalysis, and sensor applications.

Some reviews have been found regarding the use of carbon-based nanomaterials in different scientific areas. Campuzano et al. [6] studied carbon dots (CDs) and graphene carbon dots (GQDs) in electrochemical biosensing. Zhu et al. [7] debated the current state and future perspective of the photoluminescence (PL) mechanism in carbon dots (CDs), carbon quantum dots (CQDs), and polymer dots, including the four PL mechanisms: quantum confinement effect, surface state, molecule state, and crosslink-enhanced emission. Zheng et al. [8] studied carbon-based nanomaterials for biological applications, including their physicochemical properties, photostability, biocompatibility, and size. Liu et al. [9] compiled the main achievements in the past four decades, paying special attention to the principles behind synthetic chemistry, luminescence mechanisms, and applications. Tian et al. [5] compiled graphene quantum dots from chemistry to applications, including optical, electrical, and optoelectrical properties. Jana et al. [10] explored the use of CQDs for bioimaging and drug delivery in cancer. The authors explained real-time monitoring through fluorescence imaging, CD-based active and passive targeting, tumor microenvironment targeting, multifunctional targeting, in vivo imaging, biodistributions in cancer models, etc. On the other hand, many applications can be found regarding carbon quantum dots. Dall Agnol et al. [2] used spirulina-based carbon dots (bottom-up process) for stimulating agricultural plant growth. The authors studied the thermodynamic and kinetic mechanisms of the pyrolysis reaction and claimed a mean activation energy of 192.6 kJ/mol, a 10 nm average size, blue photoluminescence emission around 450 nm under a 340–400 nm excitation wavelength, and a high range of solubility when dispersed in a 0.050–1 mg/mL aqueous solution (used for lentil seed growth). Seibert et al. [11] studied the effect of different CQDs (1 wt%, 2.5 wt%, and 5 wt%) on epoxy, and the results showed an increase in the toughness of epoxy by 260%, an increase in the thermal conductivity by 144%, and an increase in the glass transition temperature by 10%, among other improvements. Sun et al. [12] studied quantum-sized carbon dots for bright and colorful photoluminescence. The authors produced CDs upon simple surface passivation, and the nanomaterials strongly showed photoluminescence in both the solution and the solid state with no blinking effect and were stable against photobleaching.

There are still some challenges regarding the mechanical properties of graphene-based nanocomposites. In spite of considerable advances in research, a lot of work must be performed in this research area. The authors seem to be unanimous on the following points: (i) scaling up the production of high-quality graphene (with the largest aspect ratio and lower number of layers) and related materials; (ii) dispersion (must not form aggregates to avoid weak points); (iii) filler/matrix bonding (to allow an efficient stress transfer mechanism); (iv) functionalization of the filler (which makes it difficult to scale up production due to excessive use of solvents and feedstock); (v) use of combined fillers to minimize some drawbacks [13]. Wang et al. [14] studied the thermal conductivity and mechanical properties of graphene nanoplatelet (GnPs)/epoxy composites containing 3 and 5 wt% GnPs using a sonification process followed by three-roll milling. The authors claimed an increase in tensile and flexural modulus but decreased strength by increasing GnP concentration. The glass transition and the thermal conductivity were also improved by GnP concentration, independently of the particle size, but the enhancement was dependent on the particle size. Prolongo et al. [15] achieved an optimum GnP loading of 8 wt% in epoxy matrix using a three-roll mill preparation method and claimed an increase of 22% in the tensile modulus, while an increase of 25% in the tensile strength was obtained (but at 3 wt% GnP). Using the same preparation method and materials, Chatterjee et al. [16] obtained an increase of 8% in the tensile modulus and 80% in the fracture toughness by using 1 wt% GnP, while Ahmadi-Moghadam et al. [17] achieved an increase of 10% in the tensile modulus by using 2 wt% GnP. Regarding graphene carbon dots, most reviews demonstrate that the applications seem to be restricted to detectors, light-emitting diodes, bioimaging, corrosion protection, or similar applications [18]. Hence, some properties are clearly affected by the particle size and fabrication method, while others are affected by filler dispersion and characteristics.

In this context, we reused the data from the already-published paper [11] using the freely available data in [19], aiming to improve the available information about CQD not related to the above applications. No additional experimental work was carried out. In our study, a profound structure–property relationship among the thermal conductivity, density, and morphological, dynamic-mechanical, and thermal properties is discussed. Also, the reinforcement mechanisms for these types of composites are elucidated.

## 2. Materials and Methods

The materials and methods are the same as in the referenced study [11] and freely available in [19]. Given this, additional experimental work was not conducted. Figure 2 shows the schematic representation of the method used for production of the nanocomposites.

The only difference was that microscopy was used to re-evaluate the surface characteristics using a 3D color map surface with projection using OriginPro version 2021. The SEM images were first converted into a matrix, and then Plot 3D colormap surface with projection. All the steps are indicated below. After the construction of the 3D colormap surface graph, the scales were standardized as 0 (the black lower limit) and 255 (the red upper limit)—this selection is made in the plot details by clicking with the right button of the mouse and selecting colormap/contours >> contour lines. It means that the higher the contrast in the original image, the higher the difference in the colormap created. For the images where the red color appears, it means that more differences in the contrast in the original images are observed. If small differences in the contrast are observed, more similarities in the color pattern are observed. Figure 3 shows a schematic representation of the 3D colormap surface graph steps.

## 3. Results

The graphene quantum dots have the behavior associated with (i) quantum confinement effects and (ii) edge effects. In other words, the behavior is associated with the dimensions of the particle [20]. In our case, since the size is constant, the discussion is carried out with respect to the dispersion of the nanoparticles and the possible changes in the crosslinking density of the epoxy.

### 3.1. Thermal Conductivity

Figure 4 shows the correlation plot (higher part) with the correlation coefficient (lower part) from Table 1 of the original study [11]. Five different variables were analyzed: mass loading (0, 1, 2.5, and 5 wt%), thermal conductivity (0.206, 0.259, 0.369, and 0.503 W/m.K), heat capacity (Cp) (1.26, 1.41, 1.38, and 1.36 J/g.K), density (0.981, 1.001, 1.005, and 0.983), and thermal diffusivity (0.167, 0.185, 0.264, and 0.376 mm^2^/s). The red color represents a positive correlation, while the blue color represents a negative correlation. Also, the more elliptic the figure, the higher the correlation. It is noted that there is a high correlation between the mass loading and the thermal conductivity (0.98) and thermal diffusivity (0.97). This is due to thermal conductivity, which refers to the ability of a given material to conduct or transfer heat, while thermal diffusivity is considered the thermal conductivity divided by the density and specific heat at constant pressure. High diffusivity means that heat transfers rapidly [21]. These results are directly related to the mass loading content due to the intrinsic characteristics of the graphene quantum dot. In addition, a high correlation between thermal conductivity and thermal diffusivity is noted (0.99).

The thermal conductivity of graphene quantum dots is dependent on the mass loading of graphene quantum dots, which is different from some graphene nanoplatelets [11,22]. It seems that there is a clear correlation between particle size and thermal conductivity. The thermal conductivity will only be affected if the particle is incorporated into the polymeric matrix. The lower particle size, the higher thermal conductivity. This occurs due to an increase in the Brownian motion and in the surface area-to-volume ratio. Hence, the thermal conductivity is expected to increase with decreasing nanoparticle size (as well as with increasing temperature and a less viscous solution). Even if an eventual agglomeration is presented, the thermal conductivity tends to increase with the CQD amount. Figure 5 shows a schematic representation of the thermal conductivity as a function of particle size and aspect ratio. By analyzing Figure 5, it is only prudent to affirm that similar thermal conductivities can be obtained even when different aspect ratios of particles are used, but for nanoparticles, the thermal conductivity is more homogeneous compared to the microcomposites. Hence, the thermal conductivity seems to be more dependent on the type of reinforcement compared to its own dispersion. It is noteworthy to mention that the effectiveness is improved with dispersion. The combination of the different nanoparticles can also increase the thermal conductivity [23,24]. It is noteworthy that the thermal conductivity of the graphene quantum dot has a high relative error (30%), and hence the measurements are performed considering the epoxy resin as the starting point material.

Many drawbacks are expected to be found when the thermal conductivity is measured in polymer matrices of nanomaterials. When the size of the object decreases to a certain value, the uncertainties of the measurement also increase, and many variables, which were not considered for fibers, for example, must be accounted for [25]. The presence of functional groups and chemical reactions tends to reduce the thermal conductivity because of the reduction in the surface area (more groups are connected to each other, and the bulk increases). Anyway, as a comparative test, the values can be considered. It is not well-known how the thermal conductivity is affected by the dispersion state of the nanoparticles. Also, knowing the aspect ratio and good contact between the different surfaces involved are also necessary for any particle [26]. Han et al. [27] compared the same volume content of reinforcement and an increase of two-fold in thermal conductivity by changing the scale from micro to nano. The authors claimed that when a percolation network is formed, more effective thermal conductivity is achieved, even for very low loadings. For conductive graphene–polymer composites, Shtein et al. [28] studied the effect of particle size on graphene–polymer composites, and the authors claimed an ultrahigh thermal conductivity of 12.4 W/m K (vs. 0.2 W/m K for neat polymers), with a percolation threshold of φ 0.17. As shown in the schematic representation in Figure 5, the thermal conductivity differs depending on the aspect ratio and size.

### 3.2. Morphological Analysis

The SEMs from the original study were reused to create a 3D color map surface with projection (Figure 6A–H) to better visualize the differences in the surface of the epoxy with the incorporation of the CQDs. 

As the CQs are incorporated into the matrix, more differences among the peaks are observed (represented by the colors red and purple). For the 2.5 wt.% CQD nanocomposite, the difference is slightly lower, while for the 5 wt.% CQD, the difference decreases considerably, being more similar to the neat epoxy. If the surface is less rough, it means that the fracture is more brittle, as in the case of neat epoxy and 5 wt.% CQDs. It can be hypothesized that for the latter, a higher amount of crosslinking occurs, since the increase in the crosslinking density leads to a more brittle material (as can be seen in the next section with the extension of the glassy plateau). The nanocomposite with 1 wt.% CQD shows a rough surface, indicating that this amount of CQD promotes more differences in the structure. Physically, a more similar “structure” leads to a narrower relaxation time distribution, and the fractal structure formed needs more energy to achieve molecular mobility, increasing the main thermal transition to high temperatures. Another analogy is that for the 5 wt.% CQD, the more homogenous structure is strongly indicative of a dominant interphase (CQD/epoxy interaction) being formed. If this is true, a lower tan delta peak has to be observed in DMTA analysis (as confirmed in the next section).

### 3.3. Dynamic-Mechanical Thermal Analysis

Figure 7 shows a representative curve of the storage modulus, loss modulus, and tan delta, representing the main transitions analyzed, and Figure 7 shows the DMTA curves of the neat epoxy and the 1 wt.%, 2.5 wt.%, and 5 wt.% nanocomposites studied. The samples were taken in triplicate, and a representative curve is shown.

The storage modulus (E’) curves can be analyzed separately in three distinct regions: the glassy region, the glass transition region, and the rubbery (elastomeric) region. Figure 7 shows the representative curve of the neat epoxy, showing the main aforementioned transitions. It is noted that the behavior is not necessarily the same when comparing the same nanocomposite. It is important to mention that the crosslinking density is not controllable; hence, it is expected that the behavior will not be the same.

Figure 8A–C shows the storage modulus, loss modulus, and tan δ curves. 

The storage modulus represents the elastic portion of the polymer. When a polymer receives an external stress (electrical, mechanical, etc.), the backbone chain deforms, aiming to dissipate the energy. The higher the energy received, the greater the deformation caused in the polymer chains. The stronger the chemical interaction, the higher the modulus, since the elastic modulus is dependent on the way the chains are packed and the intermolecular forces [30,31,32]. In the case of composites or nanocomposites, unless the reinforcement acts as a barrier to polymer chain deformation, no effect of the reinforcement is observed. For fiber-reinforced composites, the fiber size is always higher in comparison to the polymer chains, but for nanoparticle-reinforced composites, the reinforcement has a more similar size compared to the polymer chains. Hence, the stress-transferring mechanisms differ from each other. For fiber-reinforced composites, the higher dimension (length) is responsible for the tensile strength, while the extremities act as compression forces and weakening points [33,34]. Also, the lower the diameter of the fiber, the higher the efficiency of the reinforcement. For nanoparticle-reinforced composites, the reinforcement mechanism is via polymer bridging, leading to a nanoparticle network similar to colloidal structures [35]. If well-dispersed and having favorable particle–polymer interactions, it forms fractal structures via polymer bridging. A molecular dynamics simulation confirms this hypothesis [36]. Put simply, and comparing the storage modulus in the glassy state, it is noted that the incorporation of CQD did not significantly influence the elasticity of the epoxy. Hence, no reinforcement effect can be attributed to the incorporation of the CQDs. But the extension of the glassy plateau increases with CQDs. This behavior can be attributed to the fact that (i) CQDs are more thermally stable compared to the polymer, and hence a higher amount of CQDs leads to higher thermal stability, or (ii) an increase in the crosslinking density [5,37,38,39]. For lightly crosslinked systems, the increase in the crosslinking density reflects an increase in the density while for heavily crosslinked systems, the density is not altered by increasing the crosslinking density. In the case of this study, the maintenance of the density values probably indicates a heavily crosslinked system (this is the reason why the density values are not altered with CQDs), in which the CQDs help in the crosslinking formation. Considering that chemical crosslinking is already more stable than physical crosslinking, when CQDs are included and become part of the system, the system becomes more stable due to the intrinsic higher thermal stability of the nanoparticles [40,41]. Another important feature is that the rubbery region has a less abrupt decay when incorporating CQDs compared to the neat epoxy. This is also a strong indication of a more crosslinked system. Physically, when the molecular chains have enough thermal energy to considerably increase the free volume (in the glass transition region), the polymer chains are apart from each other [4,42]. If no chemical joints are formed, the loss of this property is higher compared to a system where a chemical network is formed, because this network hinders further deformation. The glass transition temperature also increases with CQD incorporation, indicative of a more crosslinked system.

The loss modulus represents the energy lost by a cycle of deformation. The energy is only dissipated when the backbone chain cannot support the energy stress imposed. The loss modulus increases by increasing molecular motion, achieving a maximum peak after a new decrease. The tan delta (the same behavior with the loss modulus, but the main events shift to higher temperatures) represents the viscoelastic behavior of the polymer (E″/E′) and represents the curve similar to the loss modulus but shifted at higher temperatures [43].

The T_g_ of the loss modulus of the neat epoxy resin is 70 °C. Lower dissipation is obtained as more CQDs are incorporated, indicating a more effective interface. In other words, when the stress is imposed through the nanocomposites by the matrix, the CQDs retain part of the energy (polymer/CQD interface), hindering the heat from dissipating and hence showing a lower peak. If a crosslinked system is presented (as is probable by the storage modulus characteristic curve), more stress can also be borne [43,44]. The peak maximum (T_g_) did not change with CQD, indicating that the segmental immobilization of the amorphous chains is similar among the CQD nanocomposites.

The glass transition of the tan delta (δ) curves was affected by the CQD content (from 70 °C for the neat resin to 120 °C for the 5 wt% CQD), followed by a significant change in the peak height (decreasing, in general). The decrease is indicative of a stronger polymer/matrix interface, causing a reduction in energy dissipation. According to Ornaghi Jr. et al. [45] and Chirayil et al. [46], the lower the tan delta peak height, the higher the constrained region amounts, representing lower molecular vibrations of the epoxy amorphous chains and a decrease in the dissipation energy. This trend is almost linear among the CQD nanocomposites [47].

Lin et al. [48] studied the origin of mechanical enhancement in polymer–nanoparticle composites with ultrahigh nanoparticle loading and claimed a value of >50%. In this particular case, a marked improvement in the mechanical properties is obtained compared to the analogous matrices. For small amounts of NPs, the mechanical properties can also be improved, but they are highly dependent on the particle size, dispersion, preparation method, and mechanical properties of the filler.

## 4. Conclusions

In this study, freely available data were reused to better understand the profound structure-versus-property relationship among the thermal conductivity, density, and morphological, dynamic-mechanical, and thermal properties of an epoxy resin with different graphene carbon dot contents (0, 1, 2.5, and 5 wt%). The main results indicate that the higher the CQD content, the higher the thermal conductivity, compared to other studies from the literature. The elastic modulus values were not altered with CQD content, but the extension of the plateau showed a considerable improvement of almost 40 °C from 2.5 wt% to 5 wt%. The mechanical properties seem not to be improved by this type of system, but other properties such as thermal conductivity and thermal electricity seem to be directly dependent on particle size and concentration.

The surfaces of the neat resin and the CQD/epoxy nanocomposites were evaluated by SEM. The 5 wt% CQD nanocomposite showed a less rough surface and a more brittle fracture, similar to the neat epoxy sample. At lower CQD concentrations (1 and 2.5 wt%), a rough surface is presented, reflecting the extension of the glassy plateau on DMTA curves.

In the glassy state, it is noted that the incorporation of CQD did not significantly influence the elasticity of the epoxy. But the extension of the glassy plateau increases with CQDs, probably due to the higher thermal stability of CQDs if compared with the polymer, and hence a higher amount of CQDs leads to higher thermal stability or an increase in the crosslinking density. Regarding the loss modulus results, lower dissipation is obtained as more CQDs are incorporated, indicating a more effective interface. The glass transition of the loss modulus and tan delta curves increases with increased CQD content, and a decreased peak height was observed due to the stronger polymer/matrix interface, causing a reduction in energy dissipation.

## Figures and Tables

**Figure 1 polymers-15-04531-f001:**
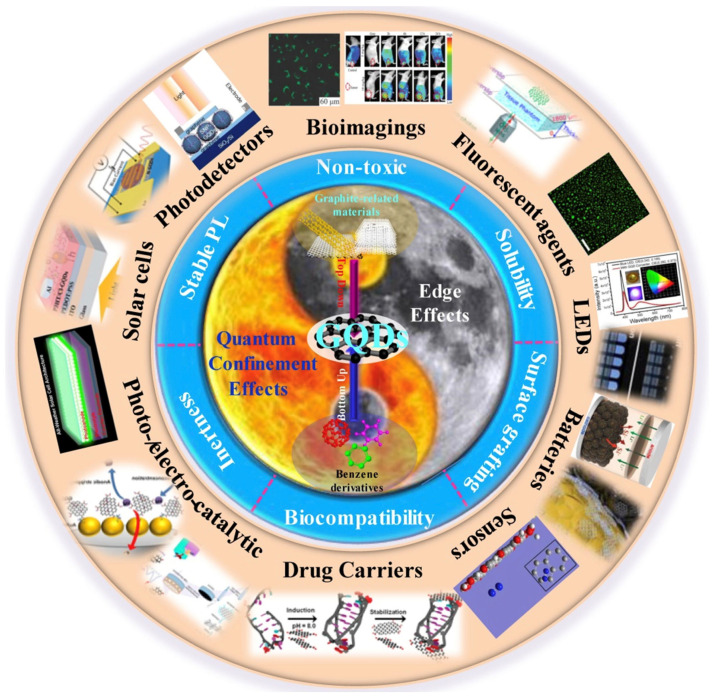
Schematic representation of the carbon quantum dot applications, properties, and fabrication. The figure was used under the Creative Commons License [5].

**Figure 2 polymers-15-04531-f002:**
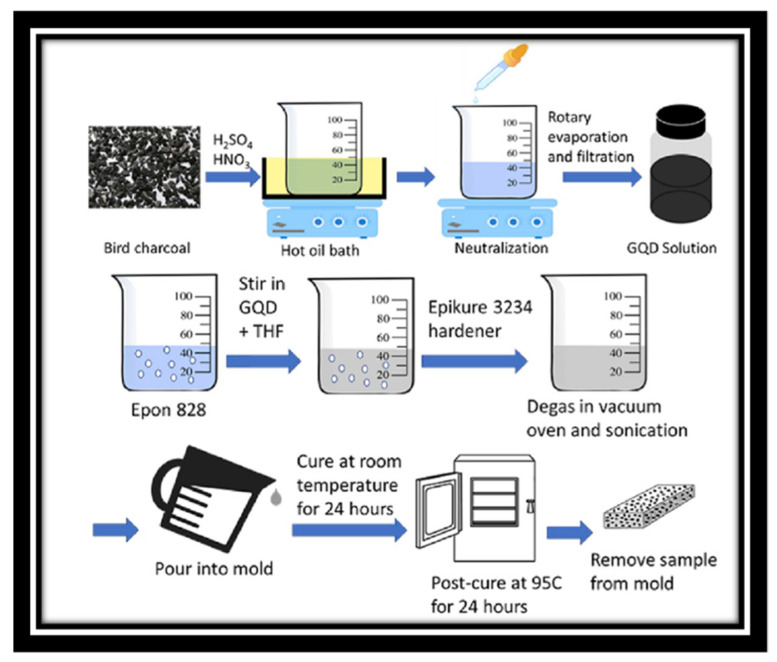
Production of the nanocomposites. The figure was used under the Creative Commons License [11].

**Figure 3 polymers-15-04531-f003:**
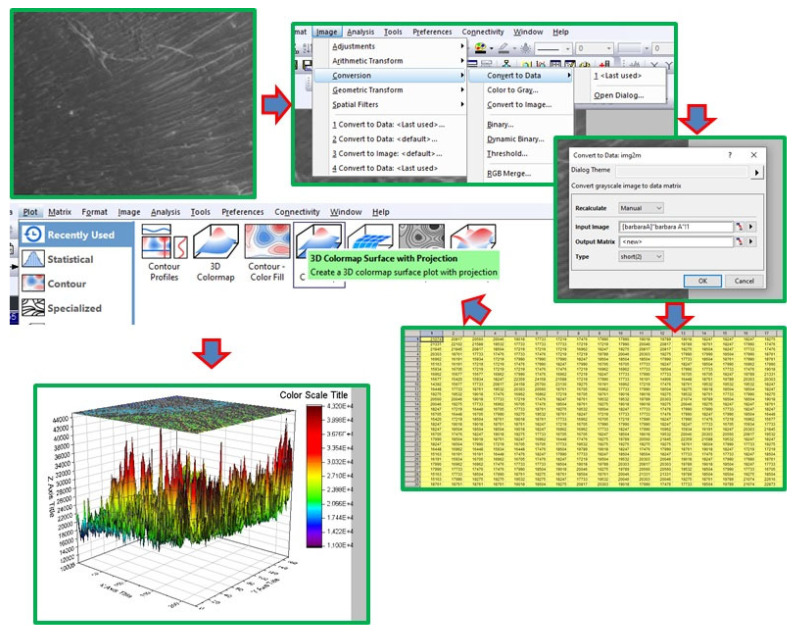
Schematic representation of the conversion of the SEM image into a 3D colormap surface with projection.

**Figure 4 polymers-15-04531-f004:**
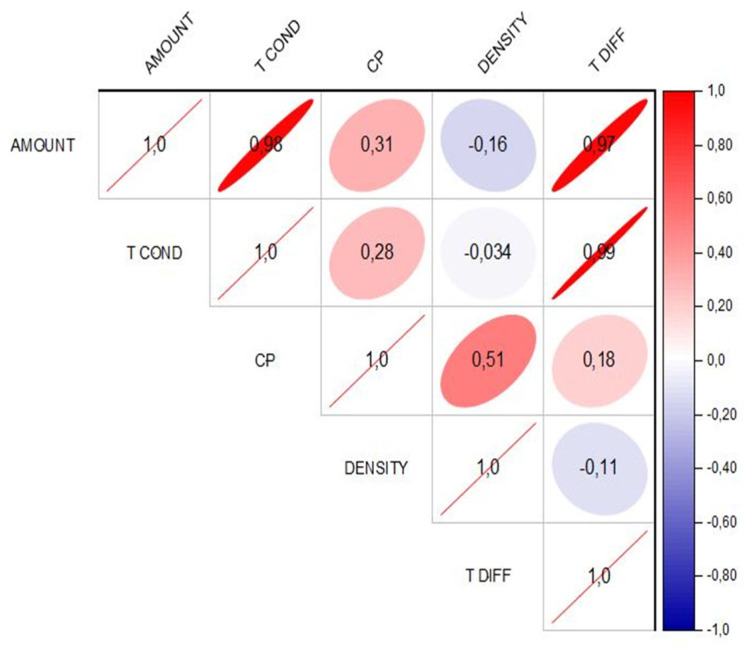
Correlation plot of the different variables studied considering the thermal conductivity. Amount—amount of CQDs, T COND—thermal conductivity, CP—heat capacity, DENSITY—density of the samples, T DIFF—thermal diffusivity. The color circles represent the positive (blue) and negative (red) correlations between different variables while the slashes (number 1.0) represent the same variable (e.g., amount vs. amount).

**Figure 5 polymers-15-04531-f005:**
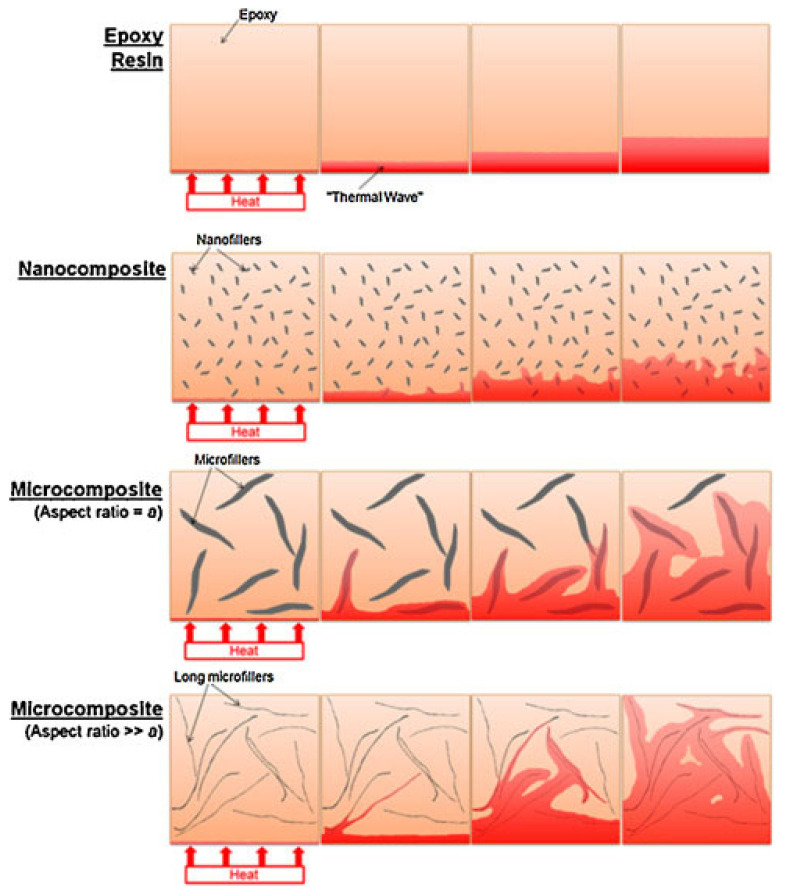
Thermal conductivity as a function of the size and aspect ratio of the nanoparticles. Figure was used with kind permission from [21].

**Figure 6 polymers-15-04531-f006:**
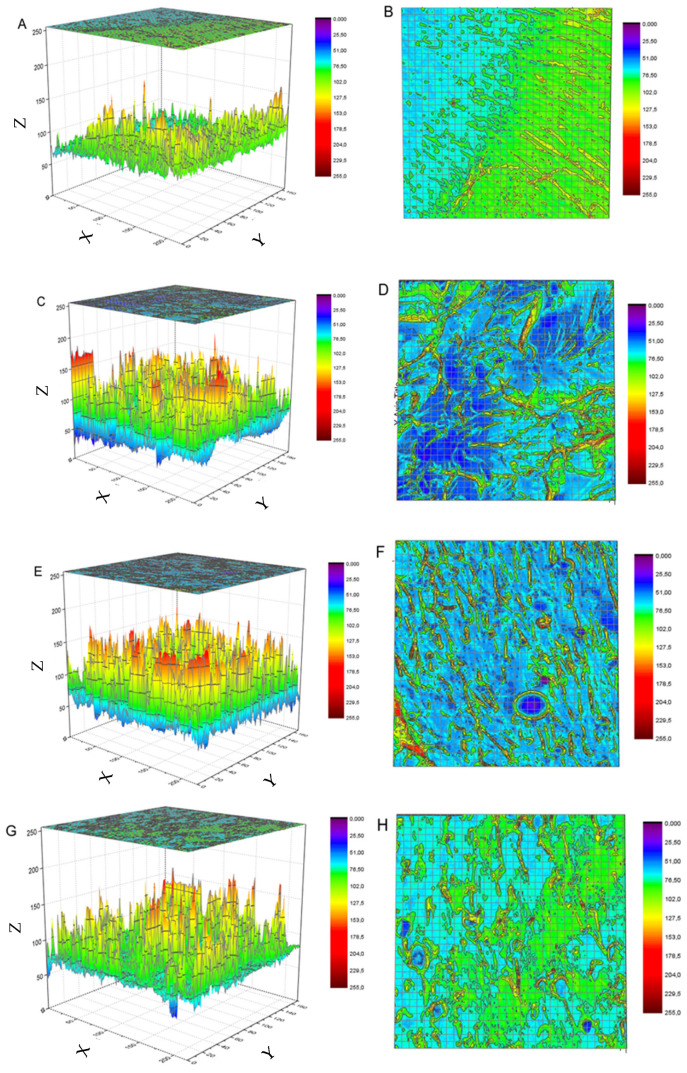
Three-dimensional color surface maps with projection with the respective upper view of (**A**,**B**) neat epoxy, (**C**,**D**) 1 wt.% CQD, (**E**,**F**) 2.5 wt.% CQD, and (**G**,**H**) 5 wt.% CQD. The original SEM images were obtained from reference [11] with kind permission.

**Figure 7 polymers-15-04531-f007:**
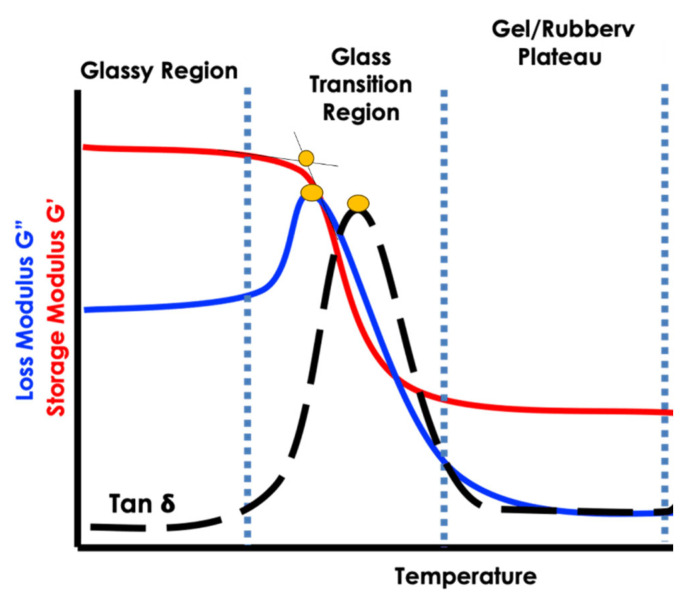
Representative curve of the storage modulus, loss modulus, and tan delta representing the main transitions analyzed. Figure was adapted from [29]. The yellow circles represent the glass transition temperature of the storage modulus, loss modulus, and tan delta.

**Figure 8 polymers-15-04531-f008:**
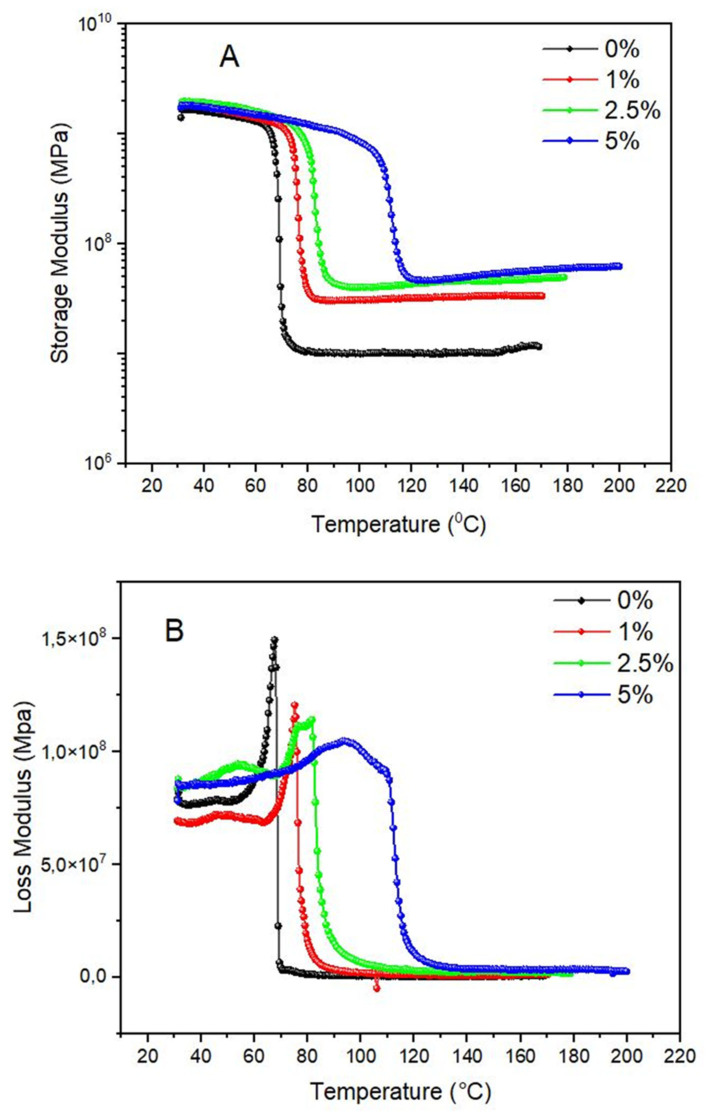
Representative curve of the storage modulus (**A**), loss modulus (**B**), and tan delta (**C**) representing the neat epoxy, 1 wt% CQDs, 2.5 wt% CQDs, and 5 wt% CQDs.

## Data Availability

Data are contained within the article.

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
