# Peer review of "Effect of the Graphene Quantum Dot Content on the Thermal, Dynamic-Mechanical, and Morphological Properties of Epoxy Resin"

_polymers, 2023, doi:10.3390/polym15234531_

Round 1

Reviewer 1 Report

Comments and Suggestions for Authors

Comments: This article reports the Effect of the graphene quantum dot content on the electrical, thermal dynamic-mechanical, thermal, and morphological properties of epoxy resin. Authors should address the following comments for its acceptance.

1.    How is the synthesis method/work different or better than those reported earlier? Author should highlight this in the introduction part.

2.    Some physicochemical characterizations are missing to confirm the structural properties for a clear understanding of readers and to enhance the manuscript quality.

3.    How did you optimize the graphene carbon dot in the epoxy composite for better thermos mechanical properties?

4.    In order to show the superiority of the current materials, comparisons over the other related materials reported in the past literatures are necessary. Mechanical and thermal properties of the current materials have to be compared with those of the other materials and reasons for performance improvements have to be discussed.

5.    The authors need to check the whole manuscript to get rid of some syntax and format errors.

Comments on the Quality of English Language

Minor spell check is required. 

Author Response

Reviewer 1

Comments: This article reports the Effect of the graphene quantum dot content on the electrical, thermal dynamic-mechanical, thermal, and morphological properties of epoxy resin. Authors should address the following comments for its acceptance.

  1. How is the synthesis method/work different or better than those reported earlier? Author should highlight this in the introduction part.

Answer: The synthesis method is the same from the reported. The results were freely available in Data in Brief Journal and re-used to corroborate the original study. This information is presented in the last paragraph of the Introduction section (highlighted in yellow).

  1. Some physicochemical characterizations are missing to confirm the structural properties for a clear understanding of readers and to enhance the manuscript quality.

Answer: Unfortunately, we do not have access to more characterizations. We-reused all the data available by the authors. Furthermore, the original article (from Polymer journal https://doi.org/10.1016/j.polymer.2019.121988) do not contains more characterizations.

  1. How did you optimize the graphene carbon dot in the epoxy composite for better thermos mechanical properties?

Answer: Since we re-use the data freely available, we do not optimize the graphene carbon (percolation threshold, etc..) and neither the authors from the original study. The information we have (and it was demonstrated in our study and in the original) is that the thermal conductivity is directly dependent on the loading mass of CQD (differently from other fillers in which a slight variation and sometimes no variation is noted with mass loading) due their small particle size allied with the confinement effects. It was constated that a more homogeneous material was obtained in using high loading masses, which reflects on a more homogeneous distribution of the fracture surface on SEM and consequently a considerably increase in the plateau of the storage modulus. The storage modulus was not affected by the CQD content because the intermolecular forces and the way the chains are packed, we not altered. Besides, the polymer chain, which dictates this property is the same for all nanocomposites and the main response of the storage modulus is attributed to the polymer phase (particles, fillers, reinforcement only affects (or not) the polymeric chain response). It was not tested high amounts of CQDs (> 5 wt.%), but it can be assumed that only the thermal conductivity response and the extension of the glassy plateau would be affected. According to Lin et al [10.1021/acs.macromol.9b02733] a considerably mechanical reinforcement is obtained for higher loading amounts (> 50 wt.%) and only if the polymer bridging mechanism is presented. Also, other authors related [http://dx.doi.org/10.1016/j.pmatsci.2017.07.004] that the reinforcement is directly related to the method process (in situ for polyurethanes, for example) or if the polymeric matrix has flexible chainbones, as PP or PE. Of course, that depending on the amount of filler, not all properties are affected - elastic modulus or maximum tensile strength are many times affected by the filler amount. We included an explanation in the results and discussion section.

  1. In order to show the superiority of the current materials, comparisons over the other related materials reported in the past literatures are necessary. Mechanical and thermal properties of the current materials have to be compared with those of the other materials and reasons for performance improvements have to be discussed.

Answer: The reviewer is correct. Other related studies were included in the discussion of the results.

  1. The authors need to check the whole manuscript to get rid of some syntax and format errors.

Answer: The whole manuscript was carefully revised. The alterations were not indicated in the manuscript.

Reviewer 2 Report

Comments and Suggestions for Authors

General:

1. Abstract: This needs to be revisited and rewritten for clarity and consistency. For example, lines 13-16 are a bit of a lengthy and unnecessary introduction that should not go in the abstract. Line 16-19 shows a very long sentence, and  Line 20-21 shows an unstructured expression as "although" is a conjunction used to contrast two statements. The abstract should also focus on research novelty and importance and how it differs from others' work or contributes to the research area.

2. Introduction:

a. This needs to be revisited and rewritten for clarity and consistency. The introduction is a bit of a literature review rather than striking a balance between briefly describing others' work and contribution, listing current challenges in the area and linking the challenges with the current aim of the research and novelty. The current aim and novelty were poorly presented in a few sentences – lines 89-92.

b. It is unclear whether the researchers used only the freely available data with no experimental work or conducted additional experimental work. In all cases, this should be clear to set the right expectations.   

3. Results and Conclusion: This needs to be revisited and rewritten for clarity and consistency. Generally, the manuscript contains long sentences; each one is more than two lines. Reading it is challenging, especially when trying to follow ideas or concepts.    

Materials and Experimental Work:

Experimental Work Section:

1. Writing: if no experimental work was conducted, then the researchers can optimize this section by referring readers to the original work to reduce unnecessary repetition. Please check and elaborate.   

Reproducibility and Repeatability

1.  Materials: There is a lack in the material description used. For instance, the supplier's name for Epon 828 and bird charcoal were not mentioned. Nothing was mentioned about the synthesis and materials for graphene carbon dot synthesis, such as which acids, their concentrations, and from which supplier. Also, nothing was mentioned about whether a hardener (i.e., curing agent) was used, and if yes, which hardener and from which supplier. Material lists and descriptions should be clearer. Please check and update all materials.

2. Sample Synthesis/Preparation Procedure and Equipment:  There is a lack of information about the procedure followed in steps not presented (e.g., whether a filtration device was used, and if yes, which device along with its specifications and for how long the filtration took place (i.e., in units of time). Similarly, if a hardener was used, at what mixing ratio? Was the degassing procedure applied? Please check and update all Sample Synthesis/Preparation Procedures and Equipment, providing details necessary for reproducibility and repeatability.

3. Software and Coding: in lines 192- 195, you mentioned, “The SEMs from the original study were re-used to create a 3D Color Map Surface with Projection (Figures 6 A-H) ….” How did you convert SEMs into 3D colour Map surface with projection, and using which software package and/or code? Please elaborate and update accordingly.  

Results:

1. Figures: some figures were not clear, and others were missed. For instance,

a.  Figure 3 does not have a legend key for the acronym used in the plot, such as T COND and  AMOUNT.

b.      Figure 4 title reads, “Thermal conductivity in function of the size and aspect ratio of the nanoparticles.” How is this relevant to the research since no caption in the result section of the manuscript was found?

c. There is no Figure 6, although the Figure 6 caption is in line 193 (Figures 6 A-H) – please notice that you did not refer to which original study (i.e., reference 12 or 13); if reference 13, Two imaging techniques were used (i.e., SEM and TEM).

d. Figure 6: it is unclear how this figure relates to the current research since no figure caption was provided in the manuscript.

Conclusion:

The conclusion section must be revisited and rewritten based on the cited above to align with the review points I highlighted above. 

Comments on the Quality of English Language

The language of the manuscript needs to be checked and improved for clarity and consistency. I have provided examples in my comments and suggestions for the authors.  

Author Response

Thank you for your email sending us the reviewers’ comments concerning our manuscript entitled " Effect of the graphene quantum dot content on the electrical, thermal dynamic-mechanical, thermal, and morphological properties of epoxy resin " (polymers-2716359 ID). All comments by the Editor and Reviewers were considered in the revised version of the manuscript and the response is included below. The modifications are highlighted in blue in the revised version. Finally, the authors would like to thank for the important recommendations which improved the readability and technical rigor of the paper. We hope that, after these modifications, the paper is suitable for publication by the Polymers journal.

Reviewer 2

General:

  1. Abstract: This needs to be revisited and rewritten for clarity and consistency. For example, lines 13-16 are a bit of a lengthy and unnecessary introduction that should not go in the abstract. Line 16-19 shows a very long sentence, and Line 20-21 shows an unstructured expression as "although" is a conjunction used to contrast two statements. The abstract should also focus on research novelty and importance and how it differs from others' work or contributes to the research area.

Answer: The reviewer is correct. The abstract was revised, and we think that now it is suitable for publication.

  1. Introduction:
  2. This needs to be revisited and rewritten for clarity and consistency. The introduction is a bit of a literature review rather than striking a balance between briefly describing others' work and contribution, listing current challenges in the area and linking the challenges with the current aim of the research and novelty. The current aim and novelty were poorly presented in a few sentences – lines 89-92.

Answer: The introduction was improved according to reviewer’s recommendation.

  1. It is unclear whether the researchers used only the freely available data with no experimental work or conducted additional experimental work. In all cases, this should be clear to set the right expectations.

Answer: We used only freely available data and re-discussed the results aiming to corroborate the former results. We highlighted this point.

  1. Results and Conclusion: This needs to be revisited and rewritten for clarity and consistency. Generally, the manuscript contains long sentences; each one is more than two lines. Reading it is challenging, especially when trying to follow ideas or concepts.

Answer: The manuscript was all revised and we hope that now it is suitable for publication.

Materials and Experimental Work:

Experimental Work Section:

  1. Writing: if no experimental work was conducted, then the researchers can optimize this section by referring readers to the original work to reduce unnecessary repetition. Please check and elaborate.

Answer: Ok. It was done. We reduced this section, and we maintained only the schematic representation of the production of the nanocomposites.

Reproducibility and Repeatability

  1. Materials: There is a lack in the material description used. For instance, the supplier's name for Epon 828 and bird charcoal were not mentioned. Nothing was mentioned about the synthesis and materials for graphene carbon dot synthesis, such as which acids, their concentrations, and from which supplier. Also, nothing was mentioned about whether a hardener (i.e., curing agent) was used, and if yes, which hardener and from which supplier. Material lists and descriptions should be clearer. Please check and update all materials.

Answer: This part was the same of the original study. We only cited the original study.

  1. Sample Synthesis/Preparation Procedure and Equipment: There is a lack of information about the procedure followed in steps not presented (e.g., whether a filtration device was used, and if yes, which device along with its specifications and for how long the filtration took place (i.e., in units of time). Similarly, if a hardener was used, at what mixing ratio? Was the degassing procedure applied? Please check and update all Sample Synthesis/Preparation Procedures and Equipment, providing details necessary for reproducibility and repeatability.

Answer: The same from above.

  1. Software and Coding: in lines 192- 195, you mentioned, “The SEMs from the original study were re-used to create a 3D Color Map Surface with Projection (Figures 6 A-H) ….” How did you convert SEMs into 3D colour Map surface with projection, and using which software package and/or code? Please elaborate and update accordingly.

Answer: More information was included in this section.

Results:

  1. Figures: some figures were not clear, and others were missed. For instance,
  2. Figure 3 does not have a legend key for the acronym used in the plot, such as T COND and AMOUNT.

Answer: A legend was included in the legend of the Figure.

  1. Figure 4 title reads, “Thermal conductivity in function of the size and aspect ratio of the nanoparticles.” How is this relevant to the research since no caption in the result section of the manuscript was found?

Answer: The Figure represents physically the results obtained by the authors from the original study. An additional paragraph was incorporated and replaced to a proper place. It was noted that the thermal conductivity increase by CQD amount by this increase could be higher if the dispersion was “ideal”, for example and considering an ideal dispersion: 5 wt% CQD should have 5 times more conductivity compared to 1 wt% CQD, but the SEM images show that the surface fracture was not homogeneous.

  1. There is no Figure 6, although the Figure 6 caption is in line 193 (Figures 6 A-H) – please notice that you did not refer to which original study (i.e., reference 12 or 13); if reference 13, Two imaging techniques were used (i.e., SEM and TEM).

Answer: The “call of figures” from the text and the figure were re-adjusted. The figures were used from reference 11 (SEM images). We properly described in the legend of Figure 5.

  1. Figure 6: it is unclear how this figure relates to the current research since no figure caption was provided in the manuscript.

Answer: Figure 6 is a schematic representation of the glass transition temperature obtained from the elastic and loss moduli and tan delta curves that will be further discussed.

Conclusion:

The conclusion section must be revisited and rewritten based on the cited above to align with the review points I highlighted above.

Answer: The conclusion was re-written.

Round 2

Reviewer 1 Report

Comments and Suggestions for Authors

The revised version of the manuscript may acceptable to the journal standard.

Author Response

Manuscript ID polymers-2716359

Status Pending minor revisions

Article type Article

Title Effect of the graphene quantum dot content on the electrical, thermal dynamic-mechanical, thermal, and morphological properties of epoxy resin

Journal Polymers

Section Polymer Composites and Nanocomposites

Special Issue Advances in Polymer/Graphene Composites and Nanocomposites

Abstract

Carbon quantum dots (CQDs) have emerged as a potential material for application in biosensing, targeted drug delivery, amongst other research fields, due to their unique properties, including fluorescence, biocompatibility, low toxicity, ease of modification, inexpensive scale-up production, and versatile conjugation with other nanoparticles. This study aimed to understand the structure versus property relationship among the thermal conductivity, density, morphological, and thermal dynamic mechanical properties of graphene carbon dot (0, 1, 2.5, 5 wt.%) epoxy nanocomposites, using the freely available data. The results showed a high correlation between mass loading, thermal conductivity, and thermal diffusivity. Although the 5 wt.% CQD nanocomposite showed a less rough surface and a more brittle fracture, comparable with the neat epoxy sample. The extension of the glassy plateau increased with higher CQDs content, as well as lower dissipation and higher glass transition temperature were observed as more CQDs were incorporated.

Keywords

Carbon quantum dots; epoxy nanocomposite; thermal conductivity; structure and property relationship

Reviewer #1

The revised version of the manuscript may acceptable to the journal standard.

Answer: Thank you for your contribution.

Reviewer 2 Report

Comments and Suggestions for Authors

1. The manuscript has improved. 

2. Since no experimental work has been conducted, the authors must provide a detailed explanation about how they get the depth dimension/information from the 2D SEM images because this is critical to the presentation of their results and conclusion. The revised manuscript must include a detailed scientific explanation for converting the 2D SEM images into matrices stating the feature-extracting mathematical algorithm applied (i.e., the SEM images are black and white, contrast and intensity need to be discussed) and then how the 2D matrices were converted into 3D ones by adding the depth dimension/information, what was the source(s) of information the authors used to get the depth dimension/information for each 2D point. OriginPro is a graphing software and, to my knowledge, can not do feature extraction from images.      

Comments on the Quality of English Language

English language needs to be further improved. 

Author Response

Manuscript ID polymers-2716359

Status Pending minor revisions

Article type Article

Title Effect of the graphene quantum dot content on the electrical, thermal dynamic-mechanical, thermal, and morphological properties of epoxy resin

Journal Polymers

Section Polymer Composites and Nanocomposites

Special Issue Advances in Polymer/Graphene Composites and Nanocomposites

Abstract

Carbon quantum dots (CQDs) have emerged as a potential material for application in biosensing, targeted drug delivery, amongst other research fields, due to their unique properties, including fluorescence, biocompatibility, low toxicity, ease of modification, inexpensive scale-up production, and versatile conjugation with other nanoparticles. This study aimed to understand the structure versus property relationship among the thermal conductivity, density, morphological, and thermal dynamic mechanical properties of graphene carbon dot (0, 1, 2.5, 5 wt.%) epoxy nanocomposites, using the freely available data. The results showed a high correlation between mass loading, thermal conductivity, and thermal diffusivity. Although the 5 wt.% CQD nanocomposite showed a less rough surface and a more brittle fracture, comparable with the neat epoxy sample. The extension of the glassy plateau increased with higher CQDs content, as well as lower dissipation and higher glass transition temperature were observed as more CQDs were incorporated.

Keywords

Carbon quantum dots; epoxy nanocomposite; thermal conductivity; structure and property relationship

Reviewer #2

  1. The manuscript has improved.

Answer: Thank you for your contribution.

  1. Since no experimental work has been conducted, the authors must provide a detailed explanation about how they get the depth dimension/information from the 2D SEM images because this is critical to the presentation of their results and conclusion. The revised manuscript must include a detailed scientific explanation for converting the 2D SEM images into matrices stating the feature-extracting mathematical algorithm applied (i.e., the SEM images are black and white, contrast and intensity need to be discussed) and then how the 2D matrices were converted into 3D ones by adding the depth dimension/information, what was the source(s) of information the authors used to get the depth dimension/information for each 2D point. OriginPro is a graphing software and, to my knowledge, can not do feature extraction from images.

Answer: This explanation was included in the manuscript in the manuscript.

  1. English language needs to be further improved.

Answer: The manuscript was carefully revised.